# Wnt7b: Is It an Important Factor in the Bone Formation Process after Calvarial Damage?

**DOI:** 10.3390/jcm12030800

**Published:** 2023-01-19

**Authors:** Bo Feng, Jun Pei, Shensheng Gu

**Affiliations:** 1Department of Endodontics, Shanghai Ninth People’s Hospital, Shanghai Jiao Tong University School of Medicine, Shanghai 200011, China; 2College of Stomatology, Shanghai Jiao Tong University, Shanghai 200125, China; 3National Center for Stomatology, Shanghai 200125, China; 4National Clinical Research Center for Oral Diseases, Shanghai 200125, China; 5Shanghai Key Laboratory of Stomatology, Shanghai 200125, China; 6Shanghai Research Institute of Stomatology, Shanghai 200125, China; 7Department of Pediatric Dentistry, Shanghai Ninth People’s Hospital, Shanghai Jiao Tong University School of Medicine, Shanghai 200011, China

**Keywords:** Wnt7b, calvarial defect model, calvarial damage, bone formation, osteogenesis

## Abstract

Objective: Previous studies found that Wnt7b played a unique and indispensable role in the process of osteoblast differentiation and could accelerate the repair of bone loss. However, what is the role of Wnt7B in osteogenesis? Is it possible to increase the expression of Wnt7b to promote the repair of skull defects? This study intends to provide the basic data for the application of Wnt7b in the treatment of craniomaxillofacial bone repair. Methods: A calvarial defect mouse model that could induce Wnt7b overexpression was established. Three days after the operation, the mice in each group were intraperitoneally injected with tamoxifen (TAM) or oil eight times every other day. There were three groups. The TAMc group (R26^Wnt7b/Wnt7b^) was injected with tamoxifen. The Oil group (3.2 kb Col1-Cre-ER^T2^; R26^Wnt7b/Wnt7b^) was injected with oil. The TAM group (3.2 kb Col1-Cre-ER^T2^; R26^Wnt7b/Wnt7b^) was injected with tamoxifen. Four weeks after the surgery, micro-CT scanning was utilized to observe new bone formation and compare the ability to form new bone around the defect area. Results: Four weeks after the operation, bone healing conditions were measured by using micro-CT scanning. The defect area of the TAM group was smaller than that of the other groups. Similarly, the bone volume fraction (BV/TV) significantly increased (*p* < 0.05), the trabecular number (Tb.N) increased, and the trabecular separation (Tb.Sp) decreased. Conclusions: Wnt7b participates in the bone formation process after calvarial damage, indicating the important role of Wnt7b in osteogenesis.

## 1. Background

Oral craniofacial bone loss is clinically very common, including periapical bone loss caused by dental pulp apical disease, periodontal bone loss caused by periodontitis, and craniofacial bone loss caused by trauma and tumor [1,2]. The treatment of bone repair involves rebuilding and maintaining new bone density to maintain normal bone function. However, in the case of large defects and systemic diseases in patients, good bone repair is often difficult to achieve [3]. Bone repair is composed of a series of coordinated biological processes, and its unique regeneration ability means it has a better damage and repair function [4,5]. Therefore, further research on the regulation mechanism of enhancing bone formation has become a key to the treatment of the abovementioned bone repair disorders [3].

Bone tissue, as a connective tissue, has a rich blood supply and is directly innervated by nerves [5]. Bone has been in a state of renewal and repair for a long time, and its unique regeneration ability is constituted by coordinated bone conduction and bone induction. Bone repair is the result of bone formation ability being greater than bone resorption. A large number of reports have suggested that WNT, TGF-β, NOTCH, and other signal pathways are involved in the regulation of bone formation [6].

In the treatment of bone defects, bone transplantation has been widely used, and bone tissue engineering provides a new option for the treatment of bone defects. Bone tissue engineering uses a three-dimensional scaffold material with good biocompatibility as a carrier, planting precursor cells and giving appropriate growth factors to promote the repair of bone defects [7]. Bone tissue engineering includes the following components: (1) a three-dimensional scaffold material, as a carrier of cells, supports the defect area and stimulates the body’s repair potential; (2) seed cells, with mesenchymal stem cells (MSCs) being currently the most widely used; and (3) bone morphogenetic proteins (BMPs) and other cytokines. BMPs have strong osteoinductive ability, which can promote bone regeneration and accelerate fracture healing. BMP2 and BMP7 have been proven to be the strongest in inducing bone formation, and they have been used in the treatment of severe clinical bone defects [8]. However, exogenous BMPs are easy to disperse and degrade after entering the body. The use of three-dimensional scaffolds can achieve a slow-release effect. It is also possible to transfect cells with genes encoding BMPs to promote high expression of BMPs. TGF-β, VEGF, and IGF-1 have been reported to be used to induce bone regeneration [9]. Cytokines are easy to dissolve and diffuse after local implantation and may be phagocytosed by macrophages. Even if gene transfection technology is used, non-viral vectors are inefficient, while viral vectors are not completely safe and the types of cells that can be transfected are limited. Although tissue engineering provides a new solution for the treatment of bone defects, a major clinical problem of bone repair is still how to promote bone repair safely and effectively.

A large number of reports have suggested that WNT, TGF-β, NOTCH, and other signal pathways are involved in the regulation of bone formation.

Gong et al. [10] discovered that WNT signal was related to bone disease in 2001, and their study found that the mutation of WNT co-receptor Lrp5 caused osteoporosis-pseudoglioma syndrome. It had been shown that the deletion of β-catenin in osteoblast progenitor cells could eliminate osteoblast differentiation, suggesting that the WNT/β-catenin signaling pathway is a key factor regulating osteoblast differentiation and might be responsible for embryonic growth. In mice, the lack of LRP5 in the whole body or in the bone could cause bone loss [11,12]. DKK1 and SOST could inhibit the transduction of the WNT signaling pathway by competitively binding receptors. Several studies found that mice lacking DKK1 or SOST showed better bone mass [13,14,15,16]. Overall, genetic evidence from humans and mice supports the importance of WNT signaling in bone formation. However, the role of the WNT signaling pathway in mediating bone formation is not fully understood [17].

Wnt7b is an important member of the Wnt family, but the research on it is limited compared to the research on other members. The effects of high expression of Wnt7b were first discovered in research studies investigating breast cancer, stomach cancer, bladder cancer, prostate cancer, and other tumors. Subsequently, more and more studies began to pay attention to its important role in embryonic development, including Wnt7b’s role in regulating the development of placenta formation, central nervous system, brain, hair, bones, joints, cartilage, blood vessels, kidneys, eyeballs, lungs, and other tissues and organs [18,19]. Li et al. [20] constructed Dermo1-Cre, a Wnt7bn/c3 transgenic mouse model by specifically knocking out Wnt7b in osteochondral precursor cells. The Wnt7b-mutant mice exhibited mineralization barriers during the embryonic stage but did not show a phenotype with a clear skeletal system after birth, and osteoblast differentiation gradually disappeared with this birth phenotype. Tu et al. [21] proposed that the above phenotype might be caused by the redundancy of WNT ligands. Chen et al. [22] found that Wnt7b, instead of Wnt5a, in the osteoblast lineage significantly enhanced the increase in bone mass caused by the increase in the number and activity of osteoblasts. Previous studies had shown that Wnt7b activated mTORC1 through PI3K-AKT signaling and partially promoted bone formation through mTORC1 activation. It was found that Wnt7b played a unique and indispensable role in the differentiation process of osteoblasts. When Wnt7b in the osteogenic precursor cell line MC3T3-E1 was knocked down, the mineralization ability significantly decreased. The mechanism was that the knockout of Wnt7b led to the downregulation of key transcription factors for osteogenic differentiation, such as RUNX2 and OSX.

The study by Chen et al. [22] found that a high expression of Wnt7b could cause an increase in bone mass, leading to an increase the number and activity of osteoblasts. However, the results of this experiment were based on the results of bone morphometry. 

Temporal genetic modification of mice using the ligand-inducible Cre/loxP system is an important technique that allows the bypass of embryonic lethal phenotypes and access to adult phenotypes. In this study, we generated a tamoxifen-inducible Cre-driver mouse strain for the purpose of widespread and temporal Cre recombination. Cre-ERT2 mice are a type of mice that express the fusion protein of a mutant estrogen receptor (ER) ligand-binding region (ERT) and Cre recombinase. Cre-ERT2 is inactive in the cytoplasm without the induction of tamoxifen; when induced by tamoxifen, the metabolite 4-OHT of tamoxifen binds to ERT, allowing Cre-ERT2 to enter the nucleus and exert Cre recombinase activity [23].

## 2. Methods

### 2.1. Experimental Animals

This experiment was carried out exactly as reported in accordance with the ARRIVE guidelines and approved by the Animal Ethics Committee of Shanghai Jiao Tong University (SH9H-2019-A193-1). The experimental animals were housed at the SPF Laboratory Animal Center of the National Clinical Research Center for Oral Diseases of Shanghai Jiao Tong University. The Col1-Cre-ER^T2^ transgenic strain was generated by fusing the cDNA of the Cre-ER^T2^ fusion protein and the downstream sequence cloned element (PA), followed by a 3.2 kb mouse type I collagen regulatory sequence [24]. Then, the 3.2 kb Col1-CRE-ER^T2^ mice were bred with R26-Wnt7b heterozygous mice to obtain 3.2 kb Col1-CRE-ER^T2^; Wnt7b heterozygous mice. Then, these mice were bred with R26-Wnt7b homozygous mice to obtain 3.2 kb Col1-CRE-ER^T2^; Wnt7b homozygous mice. Mouse toes were collected from each mouse for genetic identification [25,26]. Twelve SPF-grade, 1-month-old 3.2 kb Col1-CRE-ER^T2^; Wnt7b homozygous male mice, in groups of 6, were injected with oil (Oil group) or tamoxifen (TAM group) (10 mg·mL^−1^). Six R26-Wnt7b homozygous male mice were injected with tamoxifen (TAMc group). In summary, the three groups were as follows: The TAMc group (R26^Wnt7b/Wnt7b^) was injected with tamoxifen. The Oil group (3.2 kb Col1-Cre-ER^T2^; R26^Wnt7b/Wnt7b^) was injected with oil. The TAM group (3.2 kb Col1-Cre-ER^T2^; R26^Wnt7b / Wnt7b^) was injected with tamoxifen. The average mouse body weight was (20 ± 2) g. The drug injections started 3 days after the operation and were performed every other day (the injection dose was 10 μL·g^−1^) [27,28,29]. This experiment constructed 3.2 kb Col1-Cre-ER^T2^; R26^Wnt7b/Wnt7b^ mice and selectively induced Wnt7b overexpression in specific parts of the mice during a specific period by drugs, in order to study its function in the process of skull defect repair. The integration of data from an overexpression mouse model with a useful clinical model will offer more valuable information for skull defect repair.

### 2.2. Main Instruments and Reagents

A small electric drill (Dremel LR-077, USA), a drill bit (Diamond Burr TR12, USA), a micro-CT (Scanco viva40, Switzerland), and a qPCR instrument (ABI 9700, USA) were used.

### 2.3. Model Establishment

The method of Cooper et al. [30] was modified to establish a skull defect model. We used 1% sodium pentobarbital for intraperitoneal anesthesia (100 mg·kg^−1^). After anesthetizing the mice, we sterilized the mouse scalp with 75% ethanol twice. The scalp was cut longitudinally along the midline of the skull to expose the parietal bone, and the skull periosteum was moved with a cotton swab. A circular slice with a diameter of 5 mm was made and placed in the midline of the skull. A diamond drill bit with a diameter of 1 mm was used to grind the skull along the edge of the slice, and the bone slice was removed without damaging the dura mater and brain contents as much as possible, while the drilling area was continuously flushed with normal saline. The surgical area was washed with saline and the skin was sutured. After the operation, the mice were housed in a single cage in a clean environment at a room temperature of 22~24 °C. The mice were free to move around and had access to standard food and water. Three days after the operation, the drugs (tamoxifen or oil) were injected in the groups every other day for a total of 8 times.

### 2.4. Micro-CT Scan

A Scanco micro-CT was used for imaging 4 weeks after the operation. The entire skull with a thickness of about 1.6 mm was selected for reconstruction with 100 μ CT slices, and the threshold was 220. The skull model was reconstructed to evaluate and measure the defect area and bone mass index. The bone mass indicators included the following: (1) Bone volume fraction (BV/TV) is the percentage of trabecular bone volume in the unit of mm of the total cancellous bone volume, which is presented as the area of trabecular bone in the image as a percentage of the total area of cancellous bone. (2) Trabecular number (Tb.N) is the number of trabecular bones in units of mm, which describes the shape of the trabecular bone and represents changes in bone mass. (3) Trabecular bone separation degree (trabecular separation, Tb.Sp) is the average distance between the trabecular bones and is used to describe the trabecular bone structure; the greater the degree of separation, the greater the distance between the trabeculae and the worse the osteogenesis ability. (4) Trabecular thickness (Tb.Th) is the average thickness of the phalangeal trabecula, in um. When osteoporosis occurs, the Tb.Th value decreases.

### 2.5. Tissue RNA Extraction and Quantitative PCR

RNA was extracted from bone tissue following the method of Wang et al. [31]. The parietal bones of the three groups of mice were extracted under RNase-free conditions, quickly frozen in liquid nitrogen, and grounded into powder. Total RNA was extracted by using the TRIzol method, and reverse transcription was performed according to the TaKaRa reverse transcription kit instructions. A qPCR instrument was used to determine the mRNA expression level of Wnt7b. The expression intensity of the target gene in each sample was corrected for the intensity of its corresponding internal reference gene glyceraldehyde phosphate dehydrogenase (GAPDH), and 3 replicate wells were set up each time. The following primers were used: Wnt7b upstream primer sequence 5′-GGCCCTCGCCATTATTTTGC-3′ and downstream primer sequence 5′-CACCATTGCGTTGACCTTGG-3′, and GAPDH upstream primer sequence 5′-CACTAGGCGCTCACTGTTC-3′ and downstream primer sequence 5′-TGGTTCACACCCATGACGAA-3′. The amplification conditions were at 95 °C for 30 s, denaturation was at 95 °C for 5 s, annealing was at 57 °C for 20 s, and extension was at 72 °C for 15 s for 40 cycles, 95 °C for 30 s, 57 °C for 30 s, and 95 °C for 30 s.

### 2.6. Aniline Blue Staining

The mouse skull that had been dehydrated and decalcified was taken out, and the skeleton of the mouse skull was stained according to the instructions of the aniline blue staining kit (STN0011 Frdbio1%, phosphate method). The operation steps were as follows: ➀ Place the skull in 100% ethanol for 1 min. ➁ Dip the skull into the toluidine blue staining solution for 30 min. ➂ Rinse with tap water for 2 min, and use filter paper to absorb the water. ➃ Soak the skull in acetone until the blue-purple color is clearly visible. Aniline blue is an alkaline dye, which can clearly show the cartilage morphology, tide line, and other structures. In this experiment, it was used for skull staining to compare the size of skull defects in different treatment groups and whether there was new bone formation at the edge of the defect.

### 2.7. Statistical Analysis

GraphPad Prism 6.0 statistical software was used to perform independent sample *t*-tests for bone mass index.

## 3. Results

### 3.1. Establishment of Mouse Skull Defect Model

We made a circular slice with a diameter of 5 mm and placed in the midline of the skull. The drill head ground the skull along the edge of the slice unit and the bone was removed without damaging the dura mater and brain contents (Figure 1). After the operation, the mice could move, eat, and drink normally, and the wounds were not infected.

### 3.2. Micro-CT Analysis

Imaging with a Scanco micro-CT was performed four weeks after the operation. The entire layer of the skull was selected for reconstruction. The defect area of the TAM group was smaller than that of the Oil group and the TAMc group. The same results were obtained by staining the bone tissues with aniline blue (Figure 2).

### 3.3. QPCR Detection of Wnt7b Expression in Each Group

The expression of Wnt7b in the TAM group was significantly higher than that in the Oil group, and the difference was statistically significant (*p* < 0.01). Wnt7b expression in the TAM group was significantly higher than that in the TAMc group, and the difference was statistically significant (*p* < 0.01). There was no difference between the Oil group and the TAMc group (Figure 3).

### 3.4. Bone Mass Index Test Results

The reconstruction analysis of the defect area and the marginal area by micro-CT showed that the BV/TV of the TAM group was significantly higher than that of the Oil group and the TAMc group (*p* < 0.05): Tb.N tended to increase, Tb.Sp tended to decrease, and Tb.TH had no statistical difference (Figure 4).

## 4. Discussion

Skull defects are a common traumatic injury in clinical settings. Repair treatment is the most commonly used method in clinical operations. The repair materials used in these operations mainly include autologous materials, heterogeneous materials, and artificial bone substitute materials. Although autologous bone is the best choice for bone transplantation, there are problems such as insufficient sources of donor materials and secondary damage. Other bone replacement materials have different degrees of associated survival rates of bone grafts, immune rejection, and transmission of infection. Overall, the clinical needs of bone injuries and other defects have not been met [32,33]. At present, there are many attempts to promote the repair of skull defects, including bone tissue engineering, use of alternative materials, and bone regeneration using biological treatment technology (cell, protein, and gene therapy) [27,34], but the treatment effects are not ideal because the repair mechanism of skull defects is not completely clear. 

One point of contention in the skull bone defect model is the choice of critical size defects (CSD). CSD can be considered as the prototype of discontinuous defects, as a threshold for failed bone formation. The evaluation of CSD depends not only on the wound ruler, but also on the quality of regenerated tissue. Fibrous connective tissue migrates to the defect area more quickly, and even cover the defect area, but does not cause substantial bone regeneration; thus, the formation of new bone is the key [35].

In rats, inconsistent data on the size of CSD has been reported. Mulliken et al. [36] reported 4 mm CSD in rats. For Long Evans rats, Ray and Holloway reported 8 mm CSD. More recently, Hollinger et al. [8] confirmed that an 8 mm defect was the CSD in Long Evans and Walter Reed rats, and reported that the defect could not heal itself after 13 months, with a maximum of 10% bone formation. Aalami et al. [30] aimed to compare the bone regeneration process of juvenile and adult mice, and evaluated non-suturing-related skull defects established in the skull bones of 6-day and 60-day-old mice, with diameters of 3, 4, and 5 mm. After eight weeks of healing time, histological and radiographic evaluations showed that all three defect sizes were cut-off values in adult mice, while significant healing was observed in young mice (59%, 65%, and 44%) [37,38]. In the preliminary experiments of this study, we found that when the CSD was 3 mm, the skull defect of 2-month-old mice basically healed after 1–2 months; when the CSD was 5 mm, the skull defect of 2-month-old mice basically did not heal after two months. Thus, the CSD used in this experiment was 5 mm.

At present, conditional knockout mice, such as DSPP-Cre, Dmp1-Cre, and Oc-Cre, have been used as tools in bone and tooth related research [39]. The 3.2 kb Col1-CreER^T2^ mice, as a new type of conditionally inducible gene knockout mice, interact with the inducer tamoxifen to complete the induction process. It had been shown that the α-1 chain of type I collagen was a regulatory element of Col1a1, which could cause transgene expression in osteoblasts [24]. Although the 2.3 kb promoter element is only active in the differentiation of mouse and rat osteoblasts, the 3.2 kb promoter sequence is very active in the early stages of mouse osteoblast differentiation. Maes et al. [24] used 3.2 kb Col1/LacZ+ and Osx/LacZ+ to label mature osteoblasts and osteoblast precursor cells, respectively. During bone development, the Osx/LacZ+ labeled osteoblast precursor cells could enter the primary ossification center from the perichondrium to mediate the development of bone tissue, while the 3.2 kb Col1/LacZ+ labeled mature osteoblasts were on the surface of the perichondrium and unable to enter the ossification center to mediate bone development [40]. It was suggested that the 3.2 kb promoter sequence in 3.2 kb Col1-ER^T2^cre mice became active in the early stage of osteoblast differentiation and effectively regulated gene expression.

At present, while the mechanism of skull defect repair is not fully understood, FGFs, BMPs, Nell-1, and other genes have been reported to be involved [28]. As an important signaling pathway for bone development and bone homeostasis maintenance, Wnt signaling not only promotes bone growth but also positively regulates the healing process of long bone fractures [29]. Previous studies have shown that Wnt7b may be involved in the bone formation process after skull defects, but direct experimental evidence is lacking. In this study, Wnt7b in the TAM group was highly expressed, which was significantly different from its expression in the TAMc group and the Oil group (*p* < 0.01); however, there was no statistically significant difference in the expression of Wnt7b between the TAMc group and the Oil group, which proved that tamoxifen is not the reason for the high expression of Wnt7b. 

Bone body and fraction (BV/TV) is a common index for evaluating the bone mass of cortical bone and cancellous bone. For cancellous bone in the medullary cavity, this ratio can reflect the amount of trabecular bone mass of different samples. The increase in this value indicates that bone anabolism is greater than catabolism, and bone mass increases. In this experiment, the BV/TV of the TAM group was significantly higher than that of the TAMc group and the Oil group, suggesting a significant increase in bone mass. The trabecular number (Tb.N) and the trabecular separation (Tb.SP) can be calculated from the microstructure of the trabecular bone, which are the main indicators to evaluate the spatial morphological structure of the trabecular bone. When bone anabolism is greater than bone catabolism, Tb.N value increases and Tb.Sp value decreases. In this experiment, compared to the TAMc group and the Oil group, the Tb.N value in the TAM group increased and the Tb.SP value decreased, showing stronger osteogenic ability. In addition, we found that the defect area of the TAM group was smaller than that of the TAMc group and the Oil group, and new bone was formed at the edge of the defect. These results showed that Wnt7b was involved in the bone formation process after skull defects and promoted bone formation.

In this experiment, although we found that wnt7b might participate in the repair of skull defects, the repair area of skull defects in the experiment was relatively limited. As we mentioned, the use of bone replacement materials is becoming more and more extensive in clinical settings. We hope to find materials that not only promotes the high expression of Wnt7b but also promotes bone repair. We are committed to finding more effective ways to repair skull defects, providing the basic data for the application of Wnt7b in the treatment of bone repair.

Theoretically, if we can find a protein or a key factor that specifically regulates the Wnt7b signaling pathway, we can control and activate Wnt7b. Of course, this needs further research. However, the overexpression of Wnt7b may have some unexpected adverse effects on osteoblasts or osteoclasts, such as whether it will lead to serious imbalance between bone anabolism and bone catabolism or whether it will lead to functional changes in other tissues and organs.

In summary, this study used a mouse skull defect model to observe bone formation after a skull defect through micro-CT. The measurement is accurate, the error is small, and the results are clear. Overall, this study of the molecular mechanism of skull defect repair provides an important experimental basis for the search for ways to promote the repair of skull defects.

## Figures and Tables

**Figure 1 jcm-12-00800-f001:**
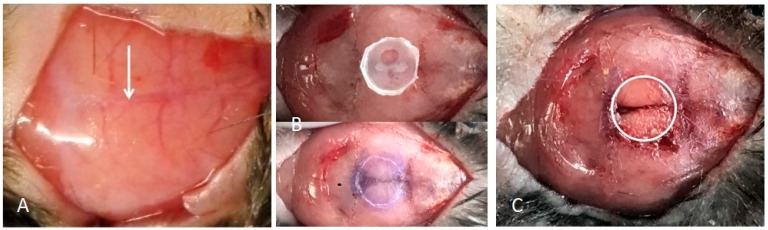
Establishment of a calvarial defect model. (**A**) Expose the operative area and find the middle cranial suture. Arrow points to middle cranial suture. (**B**) Make a circular slice with a diameter of 5 mm and place it in the midline of the skull. Grind the skull along the edge of the slice unit and the bone is removed. (**C**) Separate the bone slice and expose the defect area.

**Figure 2 jcm-12-00800-f002:**
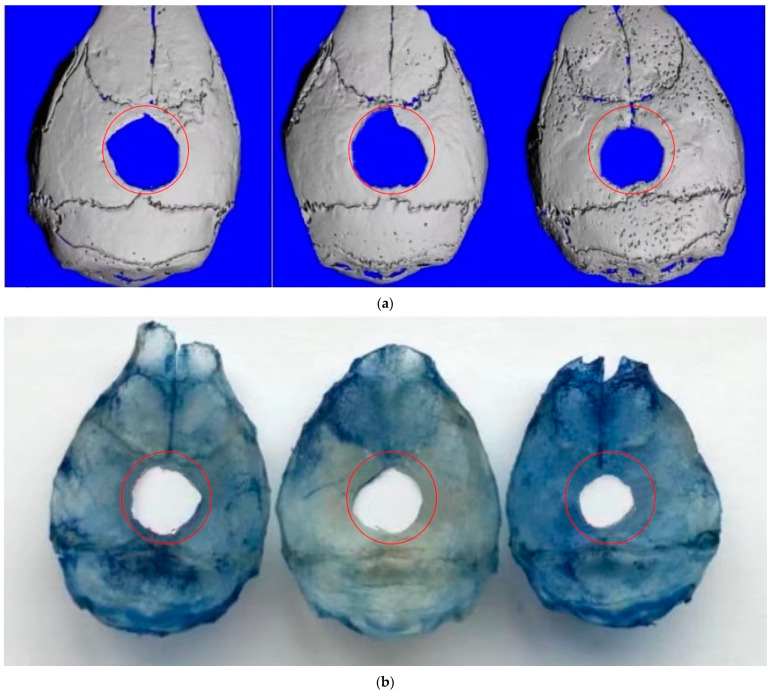
Calvarial defect between groups. The red circles mean the same size image and region of interests. (**a**) Three-dimensional reconstruction of skull by using a micro-CT. (**b**) Aniline blue skeleton staining. (Three groups: the TAMc group, the Oil group, and the TAM group).

**Figure 3 jcm-12-00800-f003:**
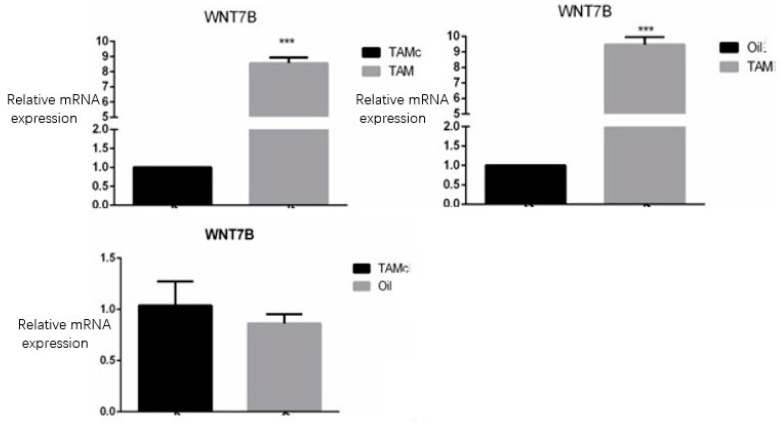
Wnt7b expression level between groups. The expression of Wnt7b in the TAM group is significantly higher than that in the Oil group (*p* < 0.01), and Wnt7b expression in the TAM group is significantly higher than that in the TAMc group (*p* < 0.01). *** means *p* < 0.01.

**Figure 4 jcm-12-00800-f004:**
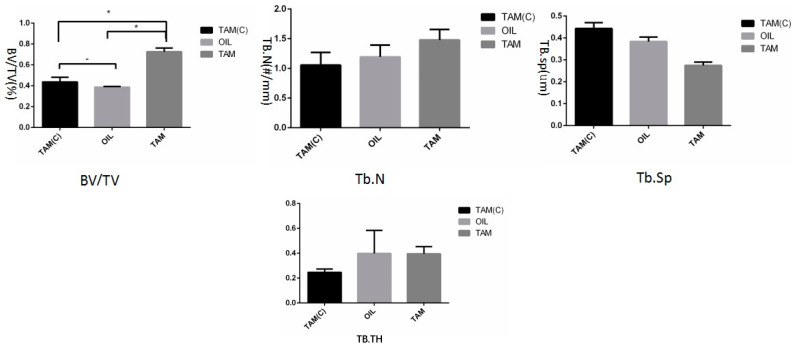
Analysis of skull bone mass index. The BV/TV of the TAM group is significantly higher than that of the Oil group and the TAMc group (*p* < 0.05): Tb.N tends to increase, Tb.Sp tends to decrease, and Tb.TH has no statistical difference. * means *p* < 0.05.

## Data Availability

The datasets generated during and/or analyzed during the current study are available from the corresponding author upon reasonable request.

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
