# Peer review of "Wnt7b: Is It an Important Factor in the Bone Formation Process after Calvarial Damage?"

_jcm, 2023, doi:10.3390/jcm12030800_

Round 1

Reviewer 1 Report

Although the manuscript is an interesting in a functional role of Wnt7b on bone regeneration in calvarial defect there are some serious problems. The authors should be addressing them.

1) Which do the authors speculate that Wnt7b has effect on the bone repair period or on regeneration volume in calvarial defect model in the present experiments? The authors should discuss them. 

2) Methods: It is unclear how to make the bone defect diameter 5mm. The authors should state the methods in detail.

Furthermore, it also is unclear that the sentence for reconstruction with 100 µ CT slices (1.6mm total). Is it reconstruction with 100 micrometer slice, or 100 slices with 16 micro meter? The authors clearly should revise the sentence.

3) Methods, section of 2.6. Aniline blue staining: Did the authors use the toluidine blue, but not aniline blue in the methods? Furthermore, the author should state what purpose is the staining.

 4)Results section Micro CT analysis: The size of ROIs were different in each image. The authors should use same size image and ROIs and compare to them.

 5) Because the group of TAMc is fewer in Tb.N and more in Tb.Sp and BV/TV than OIL it is likely to related to different value in the Tb.Th. Then the authors also should analyze the value in Tb. Th. of each group.

 6) p3 last line: Is the phrase correct ‘a diamond drill but’? Is it a mistype but, instead of bit?

Reviewer 2 Report

Dear author,

thank you for your submission. It is just a nice experiemntal animal study in mice about Wnt7b signalling.

I would recommend:

- Please correct the Abstract. It is not fully clear to everybody, what you want to tell.

- The last paragraphs of the introduction belong to the materials and methods part. Please move.

- Please re-write the discussion and tell us something about the future idea of your study. Are there therapeuticals on the way for Wnt7b activation? What adverse effects would you expect?

- Please optimize the description of the three different mouse strains.

Thank you.

Round 2

Reviewer 1 Report

The revised manuscript has improved the pointed-out problems.

The reviewer recommend the manuscript for publication.